# Evidence of Au(II) and Au(0) States in Bovine Serum Albumin-Au Nanoclusters Revealed by CW-EPR/LEPR and Peculiarities in HR-TEM/STEM Imaging

**DOI:** 10.3390/nano12091425

**Published:** 2022-04-22

**Authors:** Radek Ostruszka, Giorgio Zoppellaro, Ondřej Tomanec, Dominik Pinkas, Vlada Filimonenko, Karolína Šišková

**Affiliations:** 1Department of Experimental Physics, Faculty of Science, Palacký University, tř. 17. Listopadu 12, 77900 Olomouc, Czech Republic; radek.ostruszka@upol.cz; 2Regional Centre of Advanced Technologies and Materials, Faculty of Science, Palacký University, tř. 17. Listopadu 12, 77900 Olomouc, Czech Republic; ondrej.tomanec@upol.cz; 3Institute of Molecular Genetics of the Czech Academy of Sciences, Microscopy Centre, Electron Microscopy Core Facility, Vídeňská 1083, 14220 Prague, Czech Republic; dominik.pinkas@img.cas.cz (D.P.); vlada.filimonenko@img.cas.cz (V.F.)

**Keywords:** gold nanostructures, fluorescent nanoprobe, noble metal nanocrystal, protein nanocomposite

## Abstract

Bovine serum albumin-embedded Au nanoclusters (BSA-AuNCs) are thoroughly probed by continuous wave electron paramagnetic resonance (CW-EPR), light-induced EPR (LEPR), and sequences of microscopic investigations performed via high-resolution transmission electron microscopy (HR-TEM), scanning transmission electron microscopy (STEM), and energy dispersive X-ray analysis (EDS). To the best of our knowledge, this is the first report analyzing the BSA-AuNCs by CW-EPR/LEPR technique. Besides the presence of Au(0) and Au(I) oxidation states in BSA-AuNCs, the authors observe a significant amount of Au(II), which may result from a disproportionation event occurring within NCs: 2Au(I) → Au(II) + Au(0). Based on the LEPR experiments, and by comparing the behavior of BSA versus BSA-AuNCs under UV light irradiation (at 325 nm) during light off-on-off cycles, any energy and/or charge transfer event occurring between BSA and AuNCs during photoexcitation can be excluded. According to CW-EPR results, the Au nano assemblies within BSA-AuNCs are estimated to contain 6–8 Au units per fluorescent cluster. Direct observation of BSA-AuNCs by STEM and HR-TEM techniques confirms the presence of such diameters of gold nanoclusters in BSA-AuNCs. Moreover, in situ formation and migration of Au nanostructures are observed and evidenced after application of either a focused electron beam from HR-TEM, or an X-ray from EDS experiments.

## 1. Introduction

Fluorescent gold nanoclusters (AuNCs) embedded in bovine serum albumin (BSA) have been extensively studied in the literature since 2009, following the publication of Xie et al. [1] However, many open questions remain unsolved, especially for the witnessed dependence of the fluorescent properties with the sizes of Au nanostructures. [2,3,4,5,6,7,8,9,10,11,12,13,14,15,16,17,18,19,20,21,22,23,24,25,26,27,28,29,30,31,32,33,34,35,36,37,38,39,40,41,42]. The discussion remains controversial because many reports describe fluorescent AuNCs where the origin of such phenomenon is linked to the small size of the nanoparticles (NPs) (e.g., Burt et al. [37], Zhang and Wang [38], Zheng et al. [39]), while other reports suggest that the key to the fluorescence properties is intimately associated to the oxidation state of the gold cluster, Au(III)- [3] and/or Au(I)-complexes [40]. In Table 1, examples of selected BSA-AuNCs preparations are given, including their synthetic parameters, as reported in literature, in conjunction with the AuNCs size, oxidation state of Au when known, and maximum (maxima) of their fluorescence emissions.

From the size perspectives, AuNCs are considered intermediate systems, falling between dimensions of isolated Au atoms (0D, zero dimensional) and Au nanoparticles (3D dimensional). The fluorescent properties emerge and are tuned by the presence of discrete energy levels that mirror the nanoparticle size variance, which approach the Fermi wavelength of free electrons (e.g., [38,43]). More detailed descriptions of different types of luminescent Au nanostructures and the related emission mechanisms, including not only the free-electron model, but also surface ligand effects, are thoroughly discussed in recent reviews (e.g., [39,44,45,46]).

The oxidation state of Au is undoubtedly an important parameter to unveil for understanding the physical basis of the fluorescence phenomenon. [44] In this context, analysis of the fluorescent properties of Au clusters entrapped by proteins, such as bovine serum albumin (BSA), is particularly debated. Mostly, Au(0) and Au(I) oxidation states are observed and screened by XPS (X-ray photoelectron spectroscopy) in BSA-AuNCs [1,2,4,15,16,30,32,34,35,36]. The pseudo-polymeric structure, such as SR-(Au-SR)x- (x = 1 or 2) motif, is envisioned to be present, and from XPS results the authors [2] demonstrated that Au-Au and Au-S bonds form in the BSA-AuNCs system. It should be noted that the occurrence of the staple motif arrangement (-S-Au(I)-S-Au(I)-S-) determined in BSA-AuNCs (as in e.g., [7]) shares similarities with the arrangement observed in thiol-protected Au_25_(SR)_18_ clusters prepared by Zhu M. et al. [47,48]. Based on XRD analysis, a core-shell structure of the thiol-protected Au_25_ cluster was determined in [47]. The same group of researchers revealed the intrinsic magnetism of these thiolate-protected Au_25_ super atoms in 2009 [49]. On the contrary, the magnetic properties of BSA-AuNCs have not been investigated so far, to the best of our knowledge. Therefore, the presence of such a property represents one of the aims tackled in this study. 

Importantly, the oxidation state of Au in the final AuNCs is closely related to the experimental conditions. There are obvious differences in the presence and/or absence of a strong reduction agent in the course of AuNCs formation as shown in Table 1. In synthetic procedures where NaBH_4_ is used, BSA represents solely a template (matrix) that prevents the occurrence of coalescence processes in AuNCs at room temperature. On the contrary, when the use of NaBH_4_ is avoided (as in our previous work [41]), BSA acts as a reducing agent (probably due to tyrosine residues [1] at increased pH above its pK_a_ value) and, simultaneously, as a capping agent for AuNCs. Interestingly, even syntheses exploiting solely BSA as the reducing agent can lead to different fluorescent properties of resulting AuNCs, as shown in Table 1. The reason for such differences in fluorescent characteristics may be either the type of heating (incubator at 37 °C vs. MW heating), or the initial concentration of BSA—see Table 1. 

Several literature reports that describe organometallic complexes containing Au(I) sites show the existence of strong Au–Au bonds and the term “aurophilicity” was therefore introduced in 1989 to describe the unusual bond properties [50]. Aurophilic interaction is thought to be as strong as a hydrogen bond and shows the presence of bond length as shorter than the sum of two van der Waals radii of gold (3.80 Å). This bond characteristic is suggested to be one of the relevant factors promoting the fluorescence effect of Au(I) complexes in the UV-vis region, when investigated in the solid state [50]. The fluorescent property may then vanish in the solution, but there are scenarios where the luminescence is restored, at least for high concentrations, when a suitable solvent is used [51,52] and/or increases in the solution when AuNCs undergo self-assembly into nanoribbon structures [53]. Importantly, many Au(I) ions, being complexed with thiols and/or compounds containing other functional groups, such as carboxylates and amides, show fluorescent effects when they are partially ordered (from aurophillic interactions) [51,52]. This might also be the case of Au(I) in BSA-AuNCs. 

Concerning another oxidation state of Au (besides Au(I) and Au(0)) being evidenced in BSA-AuNCs, however implicitly (i.e., based on additional chemical reduction using NaBH_4_ and not directly by XPS), Dixon and Egusa [3] reported the presence of a significant amount of Au(III) in 2018. Terminologically, they label the systems as BSA-Au(III) complexes, but, simultaneously, they admit that there is no disproval of the existence of neutral Au(0) NCs in BSA-Au samples [3]. They [3] summarized that the red fluorescence that emerged due to Au(III) cations complexed with BSA at or above the pH value (9.7 +/− 0.2), which was responsible for the conformational change of BSA structure from normal (N) to aged (A) conformation. Therefore, they [3] concluded that upon BSA conformational change, some Cys–Cys bonds were expected to be sufficiently exposed and solvent-accessible and might interact with Au(III). They also experimentally proved multiple specific Au(III) binding sites in BSA and postulated that the UV-excitable red fluorescence of the BSA-Au(III) complex is due to the internal and potentially cascaded energy transfers among chromophores [3]. Alternatively, they admit strong electron delocalization as a source of the red fluorescence [3].

Furthermore, based on Table 1, it is obvious that sizes of AuNCs, the oxidation state of Au, and consequently their luminescent properties are strongly dependent, among others, on the exact conditions of their synthesis. Moreover, purification (if applied), storage conditions, and time elapsed before the characterization of AuNCs is completed, may represent other factors influencing the fluorescence of the final NCs as demonstrated in our previous work [41]. Obviously (Table 1), visualization of tiny AuNCs is usually performed by TEM and HR-TEM (high-resolution transmission electron microscopy). Particularly, STEM (scanning transmission electron microscopy) using High-Angle Annular Dark Field (HAADF) mode is necessary to be employed in the case of BSA-AuNCs [37]; otherwise, the TEM resolution is obscured by the enveloping protein (BSA) and the accurate size distribution cannot be obtained as stated in [2]. Interestingly, in many research papers (e.g., [3,4,42,54]), the TEM images of AuNCs exceed the size characteristic for their fluorescent properties (maximum of 2 nm in diameter). Besides HR-TEM/STEM, the determination of BSA-AuNCs sizes is often based on mass spectrometry measurements (MS) [55,56]. This is an indirect method where the MS size of BSA is subtracted from the MS size of the BSA-AuNCs system. However, this leads to the determination of the number of Au atoms per BSA. Such estimation does not provide any information about the real size of AuNCs because Au atoms are not localized on a single place in BSA; indeed, they can be spread on different cysteine and other residues of the protein [3,41]. Therefore, it seems that direct visualization of AuNCs as well as determination of individual AuNCs sizes within BSA represent challenging tasks. 

As above mentioned, the magnetic properties of BSA-AuNCs have not been investigated so far and, therefore, it is one of the aims of the present study. From the analysis of literature results [57,58,59,60,61,62,63,64,65,66], many NPs (mostly prepared using thiolate residues) comprising engineered Au(0) metal clusters give a wide range of magnetic behaviors (diamagnetic, Pauli paramagnetism, antiferro-, and ferromagnetic responses) that depend on the system size (1.8–4.4 nm), the metal’s local symmetry, metal coordination environments, and unbalanced/charged states (neutral, positive, and negative metal–organic clusters). For example, the 25 gold (Au(0)) atoms NC stabilized by 18 thiolate ligands prepared by Zhu and co-authors (Au_25_(SR)_18_, with R = phenylethyl) shows, in frozen CH_2_Cl_2_/toluene solution at *T* = 8 K an EPR (Electron Paramagnetic Resonance) signal characteristic for *S* = ½ systems with *g*-values consistent with an orthorhombic symmetry at 2.56(x), 2.36(y), and 1.82(z) [49]. Similarly, Agrachev and co-authors reported the phenylethanethiolate Au complex (neutral), Au_25_(SC_2_Ph)_18_, encoding a radius of 13.2 Å, which also exhibits EPR signatures associated to spin *S* = ½. The EPR spectrum of the complex at *T* = 5 K is rather broad (~200 mT) for Au_25_(SR)_18_ and anisotropic, with *g*-tensor components at 2.53 (x), 2.36 (y), and 1.82 (z) [67]. 

In this report, the authors used CW-EPR (Continuous Wave EPR), light-induced EPR (LEPR), HR-TEM, and STEM to thoroughly investigate the formation of BSA-AuNCs in a solution, which were intentionally prepared by Xie’s type of synthesis [1]. So far, EPR has been employed for the investigation of the emerging magnetism of atomically precise AuNCs of different sizes (e.g., Au_25_(SR)_18_, Au_133_(TBBT)_52_), which were prepared by a multistep synthesis involving NaBH_4_ as a reduction agent [5,6,49]. The EPR/LEPR study of BSA-AuNCs (further labelled as AuBSA in the present study to be directly distinguished from the other published results for BSA-AuNCs) is more complicated than the previous EPR studies of atomically precise AuNCs [5,6,49] because BSA alone reveals photochemically induced radicals’ generation, which results in damages of the BSA structure and oxidation of cysteine and tryptophan residues [68,69]. Several years ago, Lassmann and coworkers [68] analyzed, using the LEPR technique, the formation of C and S radical centers in BSA by UV-photolysis, especially thiyl (R-CH2-S•) and perthiyl radicals (R-CH2-S-S•), revealing that radiation damages are produced in the amino acid chain in a dose-dependent manner, i.e., the number of radicals increases by increasing the UV light irradiation time. The authors found that in bovine serum albumin (BSA) the thiyl radical (R-CH2-S•) exhibits axial anisotropy, with *g*_//_ = 2.17, *g*_⊥_ = 2.008, Δ*B*_pp_ = 1.8 mT, pH = 7, *T* = 80 K) and the cysteine × HCl radical (phosphate buffer, pH = 3) shows similar features, with *g*_//_ = 2.11, *g*_⊥_ = 2.011, Δ*B*_pp_ = 3.2 mT, *T* = 80 K). The *g*_//_ component was always found to be very broad and weak, to the point that it becomes barely detectable, while the *g*_⊥_ component gives the dominating, most intense signal. Together with the thiyl radicals, other types of radical species were demonstrated to form as well, which include perthiyl centres (R-CH2-S-S•) that exhibits rhombic character, and likewise, C-radical centers. In BSA (pH = 7, *T* = 80 K), EPR signals for the perthiyl specie show resonances at *g*_1_ = 2.057, *g*_2_ = 2.027, and *g*_3_ = 2.002, while the C-radical species have *g*_avg_ ~ 2.000 [68].

Therefore, the main aim of this study is to thoroughly investigate the properties of the AuBSA system with the aid of selected techniques (EPR/LEPR, HR-TEM/STEM). The following key findings are presented: (i) redox processes take place upon entrapment of Au(III) in BSA, in which the protein backbone, in absence of other reducing equivalents, provide the electrons needed for Au(III) reduction; (ii) the Au uptake process by BSA in the solution occurs on two types of sites, i.e., cysteine (Cys) and oxygen/carboxylate donor residues (e.g., tyrosine, Tyr); (iii) the magnetic interaction among cationic forms of Au arises from an admixture of Au(0), Au(I), and Au(II); and (iv) UV irradiation of the AuBSA system induces irreversible damages, and radicals are generated in BSA upon UV light irradiation. It is disclosed that no energy transfer between entrapped Au clusters and the BSA protein occurs under photoexcitation, thus the photoluminescence phenomenon is governed by the intrinsic properties of the Au system (its size and the presence of various spin active redox states) and both variables equally matter. On the other hand, in view of the extended application of these materials in biomedical scenarios, correlation between their experimental parameters (e.g., concentration in solution) and cell’s toxicity (with and upon photoexcitation) should be analyzed with great caution, because observation of an enhancement of toxic effects on cells by light irradiation might be hampered by light-induced radical’s formation on the protein itself and not solely to the photoexcited states of the gold nanoassembly.

## 2. Materials and Methods

### 2.1. Chemicals

Bovine serum albumin (>98%, BSA), gold(III) chloride trihydrate (HAuCl_4_.3H_2_O), and sodium hydroxide (NaOH) were purchased from Sigma-Aldrich (Saint Louis, MO, USA) and used as received (without any further purification) for all experiments. Deionized (DI) water prepared by purging Milli-Q purified water (Millipore Corp., Bedford, MA, USA) was used in all experiments.

### 2.2. Synthesis of AuBSA System

The procedure of our AuBSA system preparation was a slightly modified version of a method published by Xie et al. [1]. In a typical experiment, BSA solution (1 mL, 1 mM) was mixed with aqueous HAuCl_4_ solution (1 mL, 10 mM) under vigorous stirring (600 rpm). After 90 s, NaOH solution (1 M) was added to obtain a basic environment (pH ≈ 12), which induced the reduction capability of BSA. Ninety seconds later, the mixed solution was heated up in a microwave oven for 10 s (power was set to 150 W). After 2 h of ageing at room temperature, the samples were dialyzed with a 12 kDa cut-off dialysis membrane against DI water. Dialysis was performed at room temperature for 24 h, with DI water being changed twice: once after one hour and then again after the second hour. Dialyzed samples were stored in the dark at room temperature.

### 2.3. Characterization of AuBSA System

Fluorescence measurements of AuNCs were performed on a JASCO F8500 (Jasco, Tokyo, Japan) spectrofluorometer using a 1 cm quartz cuvette and 2.5 nm slits. Emission spectra were measured in the range of 500–850 nm with the data interval of 1 nm and a scan speed of 100 nm/min. The excitation wavelength was set to 480 nm. Excitation-emission 3D maps were measured in the excitation range of 250–500 nm with the data interval of 2 nm and in the emission range of 250–850 nm with the data interval of 1 nm and a scan speed of 5000 nm/min (Figure 1). All spectra were corrected to avoid any deviations of instrumental components. Samples were diluted with DI water so that the protein concentration was 2 mg/mL for emission spectra and 0.25 mg/mL for excitation-emission 3D maps.

The quantum yield of fluorescence (QY, Φ) was calculated by the following equation:(1)$$\Phi =\Phi_{s}\cdot \frac{F\cdot \left(1-10^{-A_{s}}\right)\cdot n^{2}}{F_{s}\cdot \left(1-10^{-A}\right)\cdot n_{s}^{2}}$$
where *F* is the integrated fluorescence intensity, *A* is the absorbance, *n* is the index of refraction, and subscript *s* indicates the standard. DCM, 4-(dicyanomethylene)-2-methyl-6-(4-dimethylaminostyryl)-4H-pyran, dissolved in ethanol (99,8%, Lach-Ner, Neratovice, Czech Republic) was used as a standard (Φ*_s_* = 0.437 ± 0.024) [70]. 

Absorbance was measured on a Specord 250 Plus—223G1032 (Analytik Jena, Jena, Germany) using a 1 cm quartz cuvette and double beam arrangement. As a reference, a 1 cm quartz cuvette filled with DI water was used. 

CW-EPR spectra were recorded on a JEOL JES-X-320 spectrometer (JEOL, Tokyo, Japan) operating at the X-band frequency (~9.0–9.1 GHz) equipped with a variable-temperature controller (He, N2) ES-CT470 apparatus. The cavity quality factor (Q) was kept above 6000 in all measurements. Highly pure quartz tubes were employed (Suprasil, Wilmad, ≤0.5 OD), and accuracy on g-values was obtained against the Mn(II)/MgO standard (JEOL standard). The spectra were acquired by carefully monitoring that signal saturation from the applied microwave power did not occur during signal’s acquisition. In situ light excitation EPR experiments (LEPR) were performed using a HeCd laser source operating @325 nm (max cw power of 200 mW) from Kimmon Koha Co. Ltd. (Tokyo, Japan). The UV-light was shined directly onto the sample, kept frozen inside the cavity EPR resonator, through its dedicated optical window. The light-off to light-on process was operated by an on-off light-shutter mechanism. Filling factors were kept the same (200 µL) in all experiments. Experimental parameters used for all the EPR traces shown in Figure 2A:

9.0821–9.0878 GHz frequency, 100 kHz modulation frequency, 0.03 s time constant, 2.00 mW applied power, 1.0 mT modulation width, 4 min sweep time, and 3 scans accumulated and averaged. Cavity background was subtracted. Experimental parameters for the dark to light sequential EPR spectra shown in Figure 2B,C: *T* = 90 K, 9.0802 GHz frequency, 100 kHz modulation frequency, 0.03 s time constant, 1.60 mW applied power, 0.5 mT modulation width, 30 s sweep time, and 5 min under dark followed by 20 min under light irradiation. Experimental parameters for the dark to light sequential EPR spectra shown in Figure 2D,E: *T* = 90 K, 9.0808 GHz frequency, 100 kHz modulation frequency, 0.03 s time constant, 1.60 mW applied power, 0.5 mT modulation width, 30 s sweep time, 5 min under dark conditions followed by 20 min under light irradiation, and then another 25 min under dark conditions. Experimental parameters for the EPR spectrum shown in Figure 2F: *T* = 90 K, 9.0802 GHz frequency, 100 kHz modulation frequency, 0.03 s time constant, 1.60 mW applied power, 0.5 mT modulation width, 1 min sweep time, and 5 scans accumulated and averaged. Cavity background subtracted.

HR-TEM measurements were independently performed by two different operators on two working places: (i) in the Regional Centre of Advanced Technologies and Materials (RCPTM) in Olomouc (Czech Republic) and (ii) in the Institute of Molecular Genetics (IMG) in Prague (Czech Republic). In both cases, AuBSA (the final protein concentration of 1 mg/mL) was drop-casted (2–4 µL) on glow discharged TEM copper grids covered with either a continuous 4-nm-thick carbon foil or lacey carbon and allowed them to dry spontaneously at room temperature. It is worth noting that for successful measurements, lacey carbon-copper grids (300 mesh) were used similarly as in ref. [37], and an electron beam was focused within the holes of the carbon film to obtain images of the protein-conjugated Au nanostructures without the interference of the carbon support (i.e., the measurements of samples deposited on 4-nm-thick carbon films were not successful). Then, the samples were measured by HR-TEM Titan G2 60–300 (FEI, Hillsboro, OR, USA) with an image corrector with an accelerating voltage of 300 kV in RCPTM. Images were taken with a BM UltraScan CCD camera (Gatan, Pleasanton, CA, USA). Energy Dispersive Spectrometry (EDS) was performed in STEM mode by a Super-X system with four silicon drift detectors (Bruker, Billerica, MA, USA). STEM images were taken with an HAADF detector 3000 (Fishione, Export, PA, USA). HAADF mode of STEM is intentionally used to better visualize NCs because heavier elements appear bright, while lighter elements appear dark. Imaging and data acquisition in IMG were performed with JEM-F200 TEM (JEOL, Tokyo, Japan) operated at 200 kV. TEM Images were acquired using an XF 416 CMOS camera (TVIPS, Gauting, Germany) and STEM images were acquired using the HAADF detector with a detecting angle of 24.4–89.4 mrad at a camera length set to 250 mm. EDS data were acquired using a JED 2300 X-ray spectrometer (JEOL, Tokyo, Japan) with a single 100 mm^2^ (0.98 sr) windowless SDD detector. Presented elemental maps are calculated from the raw spectra using a standardless Zeta factor method embedded in the Jeol Analysis Station software.

## 3. Results and Discussion

### 3.1. Fluorescent Properties of AuBSA

Figure 1 displays characteristic fluorescent properties of AuBSA samples. According to the 3D excitation-emission fluorescence map (Figure 1), two distinct asymmetric excitation-emission maxima (EX/EM) with bathochromic tails appeared in the ultraviolet-visible (UV-vis) spectral region: 275/345 nm (EX/EM) and 275/655 nm (EX/EM). This is in accordance with our previous work [41] as well as with the fluorescent features of NCs prepared in a similar manner (see Table 1), i.e., using BSA simultaneously as a reductant of Au(III) precursors and a template for AuNCs formation. While the former EX/EM maximum and its bathochromic tail of a low intensity (being located at around 325/400 nm EX/EM) could be attributed to Trp and oxidized Tyr residues of BSA and/or blue-emissive small AuNCs, respectively; the latter intensive EX/EM maximum can stem from Au(x)-BSA, where x represents the oxidation state of gold reaching the values of 0, I, II, (proved in this study for the first time), and III. Obviously, AuBSA can be excited by employing any wavelength in the range of 250–500 nm, as clearly demonstrated in Figure 1.

The characteristic fluorescent spectrum of AuBSA (EX 480 nm) where the center of emission (positioned at 655 nm) is marked off is shown in Appendix A for the sake of a direct comparison with fluorescence spectra of AuNCs being prepared by other authors using similar synthetic procedures (see Table 1). Namely, the visible excitations of fluorescent species serving as bio-imaging probes are preferred in practical applications because of the tissue optical window [71]. Fluorescence quantum yield (QY) of AuBSA was determined according to equation (1) and revealed the mean value of 6.7 ± 0.1% (details presented in Appendix A) with respect to DCM dissolved in ethanol.

### 3.2. CW-EPR and LEPR Experiments and Analysis

To address in more details the electronic/magnetic characteristics of AuBSA, without and under in situ UV-light irradiation (@325 nm), CW-EPR and LEPR spectroscopic techniques were employed. In addition, as reference systems, the resonance fingerprints of HAuCl_4_ (Au(III) precursor) dissolved in DI water, as well as the neat BSA protein (in DI water), were recorded under identical experimental conditions to unveil the electronic changes that might occur in the BSA protein after decoration/entrapment of Au; see Figure 2.

#### 3.2.1. CW-EPR and LEPR Analysis of Precursors (HAuCl_4_ and BSA)

Figure 2A (trace a) shows the X-band resonance signal acquired at *T* = 90 K in dark conditions of the aqueous solution of HAuCl_4_ (8.1 mg/0.2 mL). As expected, the Au^3+^ cation does not show any strong EPR resonance signals, which agrees with the diamagnetic nature of its ground state electronic configuration ([Xe]4f^14^5d^8^). Only a very weak derivative signal emerges at *g_iso_* = 2.13, which is superimposed to a broad dispersion signal extending over 100 mT that becomes visible above the background noise at a magnetic field higher than 300 mT. These weak resonances suggest that a small fraction of Au is present in the reduced forms, Au^0^ ([Xe]4f^14^5d^10^6s^1^) and/or Au^2+^ ([Xe]4f^14^5d^9^), in the hydrated Au^3+^ cations [72]. The additional resonance signal that appears in the low field region (Figure 2A, trace a), at *g* = 4.26, indicates that the reduced Au centers, Au^0^ or Au^2+^, are magnetically interacting, either via a direct or super-exchange pathway (e.g., O or OH- bridged Au^0/2+^ dimers), forming aggregates with dimensions <1 nm, and giving an effective integer spin system in which the resonance signal at *g* = 4.26 represents the half-field transition (Δ*m_s_* = 2). Upon in situ irradiation by UV-light of the frozen HAuCl_4_ aqueous solution at *T* = 90 K, no changes in the overall resonance envelope were observed (spectrum not shown). 

Different behavior is recorded for BSA in water. The EPR resonance signal observed under dark conditions is shown in Figure 2A (trace b) (*T* = 90 K, neat BSA, 7.0 mg/0.2 mL in DI water). A cluster of very weak resonances develop just above the background noise at *g_avg_* = 2.13. No other detectable high or low field resonance components are observed. These resonance signal originate from either not fully compensated cavity background noise, even after its subtraction from the sample resonance signal, or from the presence of the very minute inclusion of metal impurities in BSA, such as Cu^2+^ or low spin Fe^3+.^ During UV-light irradiation (Figure 2A, trace c) a strong signal immediately appears at *g_iso_* = 2.00, which arises from radiation damages induced in the BSA protein. These spin containing sites are based on cysteine (Cys•), tryptophan (Trp•) moieties and other C-centred radicals (C•) [68,69].

In full agreement with previous literature reports [68], the authors observed the dynamic formation of the radical species in BSA during an UV light-off (5 min) to UV light-on (20 min) EPR signal acquisition sequence, as shown in the 3D reconstructed CW-EPR/LEPR resonance plot given in Figure 2B and in the 2D CW-EPR/LEPR plot in Figure 2C. Individual CW-EPR spectra were acquired under fast-scan detection mode (30 s acquisition time for each sequential resonance spectrum). The authors noticed that formation of thiyl, tryptophan and C-centered radicals represent the early oxidation events in BSA and start to form as soon as the UV light is applied to the frozen sample kept inside the cavity resonator. The perthiyl radicals (RSS•), on the other hand, develop much later in a slower process, after application of a much longer UV irradiation time (e.g., detected after 8 min of UV irradiation in the frozen water matrix at *T* = 90 K). Figure 2F shows a well-resolved EPR resonance signal recorded for BSA in water irradiated in situ at 325 nm for 20 min at 90 K, which unveils the resonance features of the perthiyl radical (RSS•, *g_x_* = 2.052, *g_y_* = 2.018 *g_z_* = 2.002) overlapped to tryptophan-based radicals (Trp•, *g_y,z_* = 2.003, *g_x_* = 2.002, *A_N_(xx,yy,zz)* = 2.0 G, 2.0 G, 10.0 G; *A_H_**_β_*_1_*(xx,yy,zz)* = 28.3 G, 28.3 G, 28.3 G; *A_H_**_β_*_2_*(xx,yy,zz)* = 13.0 G, 13.0 G, 13.0 G). Note that the region around *g* = 2.000 clearly contains the overlapped resonance contributions of various spin species, but it is dominated by the g_y,z_ components of the thiyl radical (Cys•, *g*_y_ = 2.003, *g_z_* = 2.002) [68]. Unfortunately, the g_x_ component of this radical specie is expected to fall around 2.16–2.17 and it is known to be rather weak and broad; thus, its resonance feature remains obscured by the presence of clustered signals seen already in EPR spectrums recorded under dark conditions (see Figure 2A, trace b). Additional resonances are observed in the spectrum and originate from the hyperfine interactions of the electron spin moments of the radical species with the nuclear spin moments of hydrogen (^1^H, *I* = 1/2, natural abundance 99.98%), nitrogen (^14^N, *I* = 1, natural abundance 99.60%) and sulfur nuclei (^33^S, *I* = 3/2, natural abundance 0.76%) present in cysteine (Cys) and tryptophan (Trp) amino acid residues. For simplicity, the detailed EPR simulation parameters used for plotting the various resonance envelopes of these spin species in BSA, shown in Figure 2F, are given in the Appendix A. Therefore, a threshold of 10 min under UV-light irradiation time was used in the following analysis of AuBSA during photoexcitation, in agreement with the suggested irradiation time employed by Lassman and co-authors [68].

#### 3.2.2. CW-EPR and LEPR Analysis of AuBSA

The resonance features of the AuBSA system are much different from those observed in neat BSA. Figure 2A (trace d) shows the EPR spectrum of the AuBSA recorded under dark conditions in DI water at *T* = 90 K (7.2 mg/0.2 mL). Unlike the EPR signatures shown earlier by neat BSA recorded under dark conditions (Figure 2A, trace b), AuBSA exhibits a strong derivative signal at *g_avg_* = 2.13 (ΔB_pp_ = 13 mT) with clear shoulders developing rather symmetrically at low and high magnetic fields: one absorption component at a *g*-value of 2.47 and a derivative component at low field, with a *g*-value of 4.26. All these resonance signals arise from spin containing Au centers. The intensity of the EPR resonances coming from Au centers in AuBSA (Figure 2A, trace d) significantly increased in comparison to those seen in the EPR spectrum of HAuCl_4_ (Figure 2A, trace a); thus, a reduction process of Au(III) promoted by BSA takes place during protein incubation with the gold precursor, either to Au(I), Au(II), or to Au(0), in agreement with literature reports [1,10,16,41,73,74]. However, Au(I) is a diamagnetic cation and cannot produce the resonance features seen in the EPR spectrum. Moreover, the EPR spectrum shown in Figure 2A, trace (d), differs substantially from those reported for thiolate Au^0^ nanoclusters [5,6,49,57,58,59,60,61,62,63,64,65,66].

According to literature data, the reduction process of Au(III) by BSA leads to the formation of gold nanoparticles (AuNCs) in which the BSA protein is directly interacting with metal centers through cysteine (Cys) and methionine (Met) residues. [3,74]. Furthermore, it is also expected that Tyr residues of BSA are the moieties that provide the electron equivalents needed for driving the Au(III) reduction process [1,41], and thus, these are the amino acids that undergo oxidation. Although very weak, at *g* ~ 2.00, the presence of organic radical signals in AuBSA emerges in the EPR spectrum, but its resolution could not be improved (Figure 2A, trace d, inset). In this framework, the authors performed a set of test-experiments where either Tyr and/or Cys molecules were incubated with Au(III) salt under the same experimental conditions used in AuBSA synthesis and the resulting systems (denoted as Au-Tyr, Au-Cys) were characterized by UV-vis extinction measurements and TEM imaging. The appearance of surface plasmon resonance (SPR) peak (around 525 nm), which is characteristic for AuNPs—see Appendix A—in the Au-Tyr system confirms the following results: (i) direct reduction of Au(III) to Au(0) by Tyr and (ii) key function of the BSA protein as a matrix/template in limiting the growth of the gold nanocluster during AuBSA synthesis. No SPR was detected in the UV-vis spectrum of Au-Cys. TEM images (Appendix A) give further evidence that AuNPs, of various sizes, are formed in the Au-Tyr system. 

Based on the EPR results, it is suggested that while small, nanosized Au aggregates are indeed formed during the incubation and redox process between Au and BSA (size of Au nano-assembly estimated ~ 1 nm), an admixture of oxidation states of entrapped and clustered Au cations is indeed present. The authors speculate that the presence of both Au^2+^ and Au^0^, together with the dominant Au^+^ specie, is generated during the entrapment process of Au in BSA, in which a disproportionation event might occur in the nanocluster (NC), 2Au^+^ → Au^2+^ + Au^0^, following the initial electron delivery by BSA to Au^3+^ and reorganization of the electron distribution within the Au cluster in response to its anisotropic interactions with the BSA protein backbone. Thus, the weak signal at *g* = 2.47 and the more intense signal at *g* = 2.13 can be associated with the co-existence of two gold-based spin active species, Au^2+^ and Au^0^. Moreover, the observed EPR features cannot simply be described, in our opinion, in terms of isolated *S* = 1/2 states, but rather to an admixture of doublets and triplet states. Simulated CW-EPR spectra of AuBSA in dark conditions are shown in Appendix A. The transition observed at *g* = 4.26 may thus represent the half-field component (∆*m*_s_ = 2) of the *S* = 1 Au^0/2+^ interacting species, having for the ∆*m*_s_ = 1 transition a *g*_eff_ value of 2.13, as observed here. Assuming an axial zero-field-splitting for the high spin component of ~20.5 mT (see Figure 2A, trace d, the orange arrows indicate the estimated 2D), from a simple point dipole approximation [75], |*D*| (MHz) = 77,924 (*g*_obs_/*g_e_*)/*r*^3^, the through-space distance (*r*) between interacting *S* = 1/2 centers should fall at ~5.1 Å. The through-space distance derived from EPR analysis translates into the smallest dimension for the entrapped Au cluster in BSA of Au_6_-Au_8_ gold units. Figure 2G shows the density functional theory (DFT) calculated structures (neutral forms) [67] for an Au_6_ (triangular) and Au_8_ (4-fold edge-capped square) assemblies. From the EPR results, it is suggested that the Au(II)/Au(0) centers that are magnetically interacting to form the *S* = 1 system adopt the spin exchange coupling pathway Au(II)-Au(I)-Au(0), thus the spin containing sites Au(II)/Au(0) are located at the corners of the Au_6_ or Au_8_ nano assemblies. Such an assumption corresponds well with the early observations of Dixon and Egusa [3] who discovered that the maximum number of Au being incorporated into BSA is less than approximately 30 per BSA, but the minimum number of Au per BSA to yield red fluorescence is less than 7. 

Upon in situ irradiation of AuBSA (10 min of UV-light irradiation) at *T* = 90 K, the overall EPR signatures addressable to the Au system did not change, both in intensity and overall signal-shape. However, the clear appearance of a resonance signal around *g* = 2.00 (Figure 2A, trace e) points towards the formation of radical species, as described above, in BSA. Most importantly, these spin containing centers that formed during UV irradiation do not appear to magnetically interact with the Au NCs; in fact, the intensity of the half-field component did not increase at all, and no other additional signals arising from a new admixture of high spin species become evident in the EPR spectrum. Therefore, these BSA radical centers (Cys•, Trp•, C•) formed during photoexcitation must involve sites that are far from the amino acid (Cys) residues that directly interact with the Au nano assemblies. 

Further validation of the present findings was obtained upon following (via LEPR measurements) the in situ variation of the entire resonance line of AuBSA using an irradiation cycle of light-off to light-on and back to light-off cycle, as shown in the 3D reconstructed CW-EPR/LEPR resonance plot given in Figure 2D and in the 2D CW-EPR/LEPR plot in Figure 2E. Individual CW-EPR spectra were acquired similarly to those employed for neat BSA under fast-field scan detection mode (30 s acquisition time for each sequential resonance spectrum). The authors observed that after application of UV light, the radiation damages induced in the protein backbone in AuBSA are quickly triggered, likewise in the neat BSA system, but there is no clear impact on the EPR signals associated with the EPR signal associated to the gold species. Then, when UV light is turned off, no significant changes are observed in the radical canters of BSA (hence the protein damages are not reversible), nor changes in the EPR resonance features of the gold centers are seen. Therefore, there is no charge or energy transfer processes involved between AuNCs and the BSA protein (at 90 K), although it is a process that, sometimes, becomes observable in BSA-AuNCs, as stated in literature reports [3,9,10]. Our finding supports the results of Yamamoto et al. [40], who discussed a high photo-stability of their BSA-Au_25_NCs, albeit the authors are dealing with much smaller AuNCs (Au_6–8_) in AuBSA as derived from EPR spectra analysis.

### 3.3. HR-TEM, STEM, and EDS Analysis of AuBSA

To directly visualize our NCs in AuBSA samples, HR-TEM and STEM analysis were performed. These techniques exploit electrons that represent reduction species par excellence. Therefore, by using electrons in studies of cationic metal-organic compounds (obviously our case), irreversible transformation (i.e., reduction) may occur. Consequently, the results of visualization cannot report on the initial nanomaterial, but on the transformed one, as already known from the literature [76]. 

EDS was performed on the places of holes in the lacey-carbon Cu grid covered by an AuBSA sample, similarly as in ref. [37]. It turned out that Au is well distributed along with S elements being attributed to the protein (Figure 3a). Figure 3b,c demonstrates STEM images of the same area on the AuBSA sample prior and after the scanning of EDS for several occasions, respectively. Red circles pinpoint the areas where the changes are the most apparent. Similarly, Figure 3d,e shows the same area of AuBSA prior and after the HR-TEM measurement being performed by using a higher magnification (i.e., higher density of electron beam). White ellipses emphasize different spots on the AuBSA sample in these two images—see Figure 3d,e. The authors can sum up that Au nanoparticles (NPs, sizes exceeding 2 nm) are formed by the focused electron beam if working with a higher magnification of HR-TEM and/or if using EDS mapping. Our observation can be a clue and indirectly confirms why there are discrepancies in sizes of fluorescent NCs presented in the literature (e.g., [3,4,42,54]) and summarized in Table 1. Another set of STEM images taken on the same AuBSA sample using a different HR-TEM/STEM machine (located at IMG in Prague) is shown in Appendix A. Independently, these additional measurements confirmed that it is possible to visualize NCs of sizes well below 2 nm without inducing their size modification. Moreover, based on TEM, HR-TEM, and STEM images, particle size distributions (PSD) were determined, and appropriate histograms are shown in Appendix A. The average AuNC sizes of 1.01 ± 0.24 nm and 1.18 ± 0.21 nm were derived from TEM and STEM images of AuBSA samples, respectively. On the other hand, the average AuNC size in the same AuBSA sample increased to 2.28 ± 0.63 nm after HR-TEM and/or EDS measurements. This corroborates the above-discussed observation about NPs in situ formation. Lattice planes of Au(111) and Au(200) were detected in NCs of AuBSA sample in a particular HR-TEM image as provided in Appendix A. 

Furthermore, HR-TEM and STEM images of the same spot on the AuBSA sample are compared in Figure 3f,g. Indeed, the HR-TEM image (Figure 3f) was taken first, followed by STEM imaging (being recorded for a better contrast, Figure 3g). Red circles demonstrate the shifts of Au nanostructures, which migrated right after HR-TEM imaging. The observed Au NPs formation and their further migration upon HR-TEM imaging can be explained by considering the cationic state of Au in AuBSA (as determined by EPR measurements here) and reduction ability of electrons employed during TEM imaging. Indeed, Au(I) as well as Au(II) being attached to BSA, most probably through sulfur atoms (as evidenced by EPR and EDS in this study), can be reduced to Au(0). Since this Au(0) may be less strongly connected with the protein backbone than cationic Au in AuBSA, it can migrate on the carbon support of the TEM grid (which is more pronounced in cases of lacey-carbon rather than on 4-nm carbon coating of the Cu grid). By meeting other Au(0) atoms, they can coalesce and form larger Au nanostructures than NCs, i.e., NPs as evidenced in Figure 3c,e,g, and confirmed also by an increased value of average AuNC size (2.28 ± 0.63 nm). The above-mentioned observation of Au NPs formation and their migration during HR-TEM imaging could also explain why the difference in size of red and blue emitting systems in ref. [4] was not as clear as expected. Logically, this supports the idea that AuNCs samples might be influenced by the interaction with the focused electron beam and the resulting images do not show the real structure of AuNCs. In other words, the presence of any cationic Au center (Au^+^ and/or Au^2+^ and/or Au^3+^) in BSA-AuNCs render such systems as being much more sensitive to techniques that use high energy electrons and/or X-ray irradiation and can explain the size and oxidation state discrepancies of Au nanoassembly in BSA-AuNCs found within the scientific literature (compared in Table 1). 

Based on our results, the authors would like to emphasize that results strongly depend on the type of experimental techniques employed for AuNCs characterization and subsequent data interpretation. Many powerful experimental techniques frequently used for nanomaterials’ characterization exploit electrons and/or are working with strong irradiation (X ray), thus affecting the electronic structure of the studied material, e.g., TEM, XAS, EDS, etc., respectively. On the contrary, EPR represents a mild technique which uses the magnetic field and microwave irradiation and, thus, can be envisaged as a very powerful technique suitable for probing AuBSA systems in particular, as well as BSA-AuNCs in general, because it does not physically alter the oxidation state of the nanomaterial in the course of data acquisition. Moreover, by using LEPR, any changes in the electronic structure of a particular nanomaterial exposed to electromagnetic irradiation (UV in our case here) can be investigated and evidenced in situ.

## 4. Conclusions

The present work demonstrates that a redox process that leads to an admixture of Au(0), Au(I), and Au(II) is taking place upon entrapment of Au(III) in BSA and several Au atoms of AuNCs are directly interacting through Cys and Tyr residues with BSA. The authors observed that Tyr molecules are able to reduce Au(III) to Au(0) in the form of AuNPs, while Cys did not provide the electron equivalents needed for the Au(III) reduction. It is also demonstrated that UV irradiation induces irreversible damages in AuBSA and radical species are generated during photoexcitation in the protein backbone. Since there is no evidence of electronic communication between AuNCs and BSA according to the LEPR experiments (in situ off-on-off cycles under UV light, at 90 K), any energy and/or charge transfer can be excluded as a key mechanism for expressing the bright fluorescence phenomenon observed, with maximum emission at 655 nm, in AuBSA. Furthermore, evidence was given for the in situ growth of AuNCs to AuNPs and migration of Au nanostructures during HR-TEM and EDS data acquisition. This work further reinforces the knowledge that results obtained by using sophisticated techniques exploiting electrons and/or X-ray irradiation in the investigation of gold nanostructures embedded in protein matrixes must be considered carefully, because the experimental evidence might be severely hampered by induced radiation damages on the protein structure.

## Figures and Tables

**Figure 1 nanomaterials-12-01425-f001:**
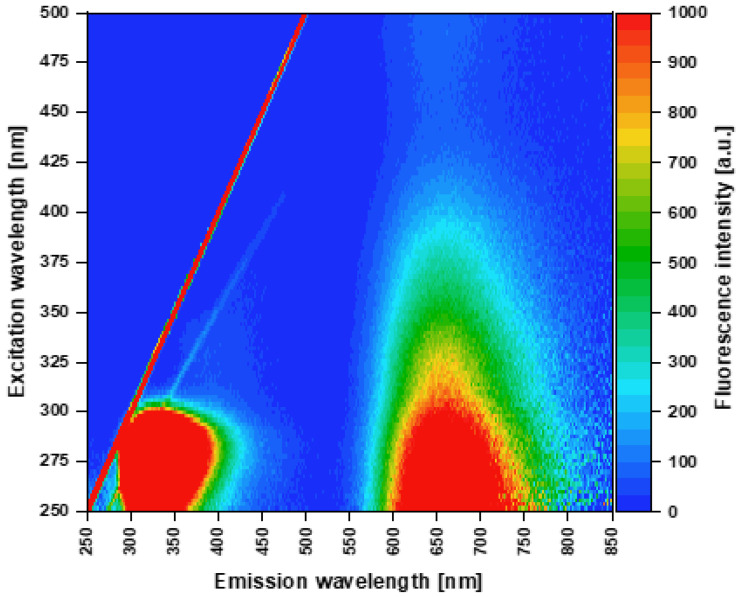
Fluorescence 3D excitation-emission map of AuBSA. Note: Strong Rayleigh (red line) and weak Raman (blue line) scattering first-order maxima are visible as straight lines in the map and represent artifacts that do not belong to AuBSA.

**Figure 2 nanomaterials-12-01425-f002:**
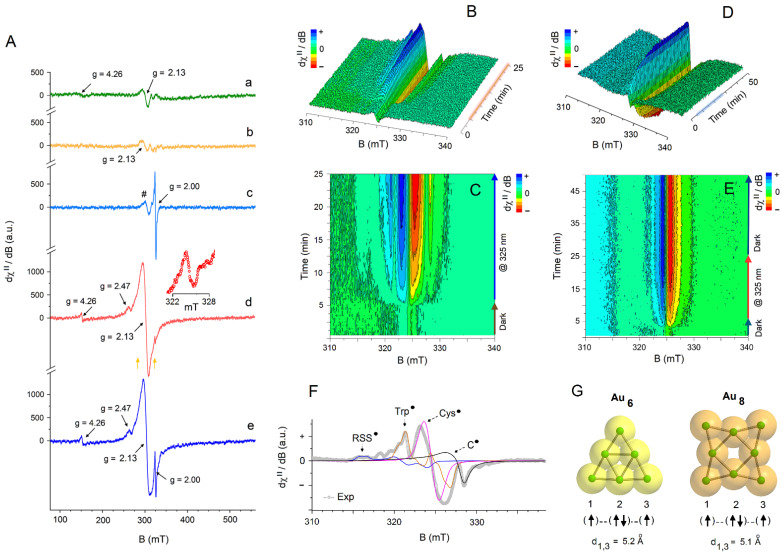
Panel (**A**) CW X-band Electron Paramagnetic Resonance (EPR) spectra recorded in water at *T* = 90 K under dark conditions of (a) HAuCl_4_, (b) BSA, and (d) AuBSA. The LEPR spectra obtained under in situ UV irradiation (@325 nm, for 10 min) of the frozen matrix solutions for the various samples (*T* = 90 K) are given as trace (c) for BSA and trace (e) for AuBSA. Note that in trace (c), the symbol (#) indicates the region in which the g_x_ component of the thyil radical (Cys•) in BSA is expected to be observed. The in situ 3D/2D LEPR experiment (*T* = 90 K, water) for BSA is shown in panel (**B**) and panel (**C**); panel (**D**) and panel (**E**) show the 3D/2D LEPR experiment for AuBSA. Panel (**F**) gives the experimental EPR spectrum (*T* = 90 K) together with computer simulations of the individual spin components obtained for neat BSA in water after 20 min of continuous UV irradiation. Panel (**G**) shows the structural organization of gold nanoclusters (Au_6_ and Au_8_) derived from DFT theory (neutral forms, coordinate files taken from ref. [67]) encoding the minimum dimensions for the Au nano-assemblies in BSA that are compatible with the distances calculated by EPR for a pair of spin–spin interacting Au(II)/Au(0) centers (labelled as 1,3).

**Figure 3 nanomaterials-12-01425-f003:**
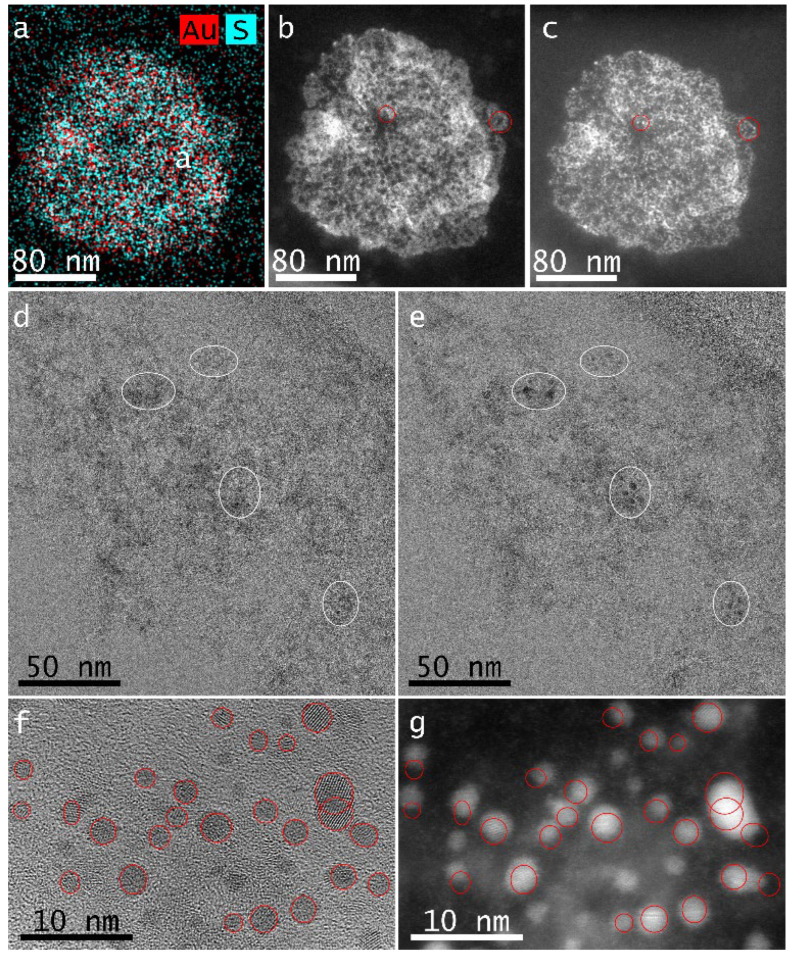
(**a**) Elemental map of Au and S distributions in AuBSA determined by EDS. STEM images of AuBSA prior (**b**) and after (**c**) the performed EDS analysis. HR-TEM images of the same area of AuBSA prior (**d**) and after (**e**) the exploitation of a higher magnification. HR-TEM (**f**) and STEM (**g**) images of the same spot on AuBSA. Colored and/or white circles indicate appearance of new NPs in (**b**–**e**) and/or migration of NPs in (**f**,**g**).

**Table 1 nanomaterials-12-01425-t001:** Details of selected BSA-AuNCs preparations and properties as reported in literature.

Reference Number	Oxidation State of Au	Size of BSA-AuNCs [nm]	Emission Wavelength Maximum [nm]	Quantum Yield [%]	Synthesis Conditions (X = BSA + HAuCl4)
[1]	Au^0^, Au^+^ XPS	≈0.8 TEM	640	6	X –(2 min)> NaOH → incubation at 37 °C for 12 h
[2]	n.a.	≈1 TEM	n.a.	n.a.	X → NaBH_4_ → incubation at RT for 1 h
[3]	n.a.	6.3 ± 2.9 (pH 12) * 3.3 ± 1.4 (pH 10) * 1.6 ± 0.7 (pH 9) *	640 (pH 12) n.a. (pH 10) 440 (pH 9)	n.a.	same as [1], NaBH_4_ used in later steps
[4]	Au^0^, Au^+^ XPS	4.2 ± 0.5 TEM	676	4.14	X → NaOH → MW (incubation at 80 °C for 4 min)
[4]	Au^0^ XPS + FQ	3.1 ± 0.4 TEM	436	1.94	X → NaOH → MW (incubation at 135 °C for 4 min)
[7]	n.a.	n.a.	640, 710	n.a.	X → incubation at 37 °C overnight
[15]	Au^0^, Au^+^ XPS	2.1 ± 0.3 TEM	650	1.9	X → NaOH → MW (120 W, 2 min)
[16]	Au^0^, Au^+^ XPS	n.a.	656	n.a.	same as [1]
[27]	n.a.	n.a.	705	n.a.	X –(1 h) > NaOH → incubation at 45 °C for 4 h
[30]	Au^0^, Au^+^ XPS	n.a.	635	n.a.	same as [1]
[32]	Au^0^, Au^+^ XPS	4–6 TEM	650	≈8	X –(30 min)> NaOH → incubation at 50 °C for 3–4 h
[34]	Au^0^, Au^+^ XPS	n.a.	685	≈5.5	X → ascorbic acid → NaOH → incubation at 37 °C for 5 h
[35]	Au^0^, Au^+^ XPS	1.6 HR-TEM	604	n.a.	X –(2 min)> NaOH → MW and then incubation at 37 °C for 12 h X –(10 min)> NaOH → incubation at 37 °C for 24 h
[36]	Au^0^, Au^+^ XPS	n.a.	620	n.a.	X → NaOH → incubation at 37 °C for 12 h + dialysis for 48 h
[37]	n.a.	average < 2 TEM 1.5–1.8 HAADF-STEM	n.a.	n.a.	X → NaBH_4_ → incubation for 3 h
[41]	n.a.	3.6 DLS	650	6.2	X –(90 s)> NaOH → MW (150 W, 10 s)
[42]	n.a.	≈5 TEM, HR-TEM	645	n.a.	X → NaOH → MW (300 W, 6 min)

Notes: n.a. = not available, * = sizes obtained after the reduction performed by addition of NaBH_4_ to Au(III)-BSA systems which were generated at given pH values, MW = microwave radiation heating, XPS = X-ray photoelectron spectroscopy, FQ = fluorescence quenching, TEM = transmission electron microscopy, HAADF-STEM = high-angle annular dark-field scanning transmission electron microscopy, HR-TEM = high resolution transmission electron microscopy.

## Data Availability

Not applicable.

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
