# Peer review of "Evidence of Au(II) and Au(0) States in Bovine Serum Albumin-Au Nanoclusters Revealed by CW-EPR/LEPR and Peculiarities in HR-TEM/STEM Imaging"

_nanomaterials, 2022, doi:10.3390/nano12091425_

Round 1
Reviewer 1 Report
The authors present an interesting EPR study of Au nanocluster-bovine serum albumin (AuBSA) materials that show, for the first time, that both Au^+2 and Au^0 oxidation states are present. The results testify to the utility of EPR for characterizing Au-organic materials with small AuNCs that can be damaged by electron imaging. The work is of broad interest for these types of biologically relevant Au-based nanomaterials and should be published. I have just a few comments that would improve the clarity of the manuscript:
The Introduction provides a very thorough background on the current state of knowledge regarding fluorescence in AuBSA materials. However, this section is quite long and provides more details on previous literature than needed. Moreover, the discussion of different synthesis methods on p. 3 is rather confusing. It might help to have a table or schematic summarizing synthesis methods and fluorescence observations so that the key previous findings are clear. In addition, the switch from AuBSA to EPR spectroscopy on the top of p. 4 is rather abrupt and the connection is not immediately clear, especially since it wasn't clear until reading to the end of p. 4 that this work appears to be the first EPR study on AuBSA. I recommend streamlining the Introduction to make it shorter and easier to follow.
The caption of Figure 2 is excessively long. I recommend putting all of the acquisition details in the Experimental section. Also, the advantage of showing both 3D plots as panels B/D and 2D plots as panels C/E is not clear. To me, the 2D plots in C/E more clearly show the dynamics. Finally, why are the colors denoting +/- values on panels B/C and D/E opposite? It is confusing.
I found the reasoning leading to the conclusion that the AuNCs mostly consist of 6-8 Au units (lines 446-450) is difficult to follow. It would be helpful for non-experts in EPR to have a more thorough explanation of how the conclusion of the AuNC size was reached.
Reviewer 2 Report
To my knowledge, this is the first report analyzing the BSA- AuNCs by CW-EPR/LEPR technique. In addition to the presence of Au(0) and Au(I) oxidation states in BSA-AuNCs, the authors show the presence of a significant amount of Au(II), which may be due to the disproportionation 2Au(I) → Au(II) + Au(0).
In this work, it is clearly shown that the redox process leading to the formation of Au(0), Au(I), Au(II) occurs when Au(III) is captured in BSA and tyrosine molecules are able to reduce Au(III ) to Au( 0) in the AuNP form, while cysteine is not.
The manuscript under review is well written. The topic was adequately presented and discussed. In my humble opinion, this manuscript can be published.
Reviewer 3 Report
The authors have prepared BSA-AuNCs and then BSA-AuNCs have been investigated by CW-EPR, LEPR, HR-TEM, STEM, and EDS. The authors have demonstrated BSA-AuNCs composed by Au(0) and Au(I) oxidation states. Moreover, the authors have observed the mechanism based on NCs: 2Au(I) -> Au(II) + Au(0). In situ formation and migration of Au nanostructures have been observed and evidenced via HR-TEM, or X-ray from EDS experiments. Overall, this work can inspire more understanding of Au nanostructures. Therefore, I would like to recommend this work to publish in Nanomaterials. Below are a few suggestions for the authors.
1. Please check the abbreviation of bovine serum albumin – embedded Au nanoclusters. (Au-BSA or AuBSA?) Please use only one abbreviation for one material.
2. For HR-TEM of Figure 3f, the lattice plane of Au-BSA should be provided to reveal the formation of Au nanoclusters in the manuscript.
3. This paper would be more impressive if the average size of Au-BSA could be calculated and provided in this manuscript based on their HR-TEM image.
4. For the introduction “However, many open questions remain unsolved, especially for the witnessed dependence of the fluorescent properties with the sizes of Au nanostructures”, more references could be cited to broaden the introduction.
https://doi.org/10.1021/acssuschemeng.9b03048
